# Mechanism of Phase Separation in Aqueous Two-Phase Systems

**DOI:** 10.3390/ijms232214366

**Published:** 2022-11-19

**Authors:** Amber R. Titus, Pedro P. Madeira, Luisa A. Ferreira, Vladimir Y. Chernyak, Vladimir N. Uversky, Boris Y. Zaslavsky

**Affiliations:** 1Cleveland Diagnostics, 3615 Superior Ave., Cleveland, OH 44114, USA; 2Centro de Investigacao em Materiais Ceramicos e Compositos, Department of Chemistry, 3810-193 Aveiro, Portugal; 3Department of Chemistry, Wayne State University, 5101 Cass Avenue, Detroit, MI 48202, USA; 4Department of Mathematics, Wayne State University, 656 W. Kirby, Detroit, MI 48202, USA; 5Department of Molecular Medicine and Byrd Alzheimer’s Research Institute, Morsani College of Medicine, University of South Florida, Tampa, FL 33613, USA

**Keywords:** Fourier Transform Infrared spectroscopy, dynamic light scattering, water structure, hydrogen bonds, phase separation

## Abstract

Liquid-liquid phase separation underlies the formation of membrane-less organelles inside living cells. The mechanism of this process can be examined using simple aqueous mixtures of two or more solutes, which are able to phase separate at specific concentration thresholds. This work presents the first experimental evidence that mesoscopic changes precede visually detected macroscopic phase separation in aqueous mixtures of two polymers and a single polymer and salt. Dynamic light scattering (DLS) analysis indicates the formation of mesoscopic polymer agglomerates in these systems. These agglomerates increase in size with increasing polymer concentrations prior to visual phase separation. Such mesoscopic changes are paralleled by changes in water structure as evidenced by Attenuated Total Reflection—Fourier Transform Infrared (ATR-FTIR) spectroscopic analysis of OH-stretch bands. Through OH-stretch band analysis, we obtain quantitative estimates of the relative fractions of four subpopulations of water structures coexisting in aqueous solutions. These estimates indicate that abrupt changes in hydrogen bond arrangement take place at concentrations below the threshold of macroscopic phase separation. We used these experimental observations to develop a model of phase separation in aqueous media.

## 1. Introduction

The importance of liquid-liquid phase separation (LLPS) in the organization and function of cells has become increasingly recognized [1,2,3,4]. LLPS is a ubiquitous process that drives the formation of membrane-less intracellular organelles, including but not limited to stress granules, nucleolus, centrosomes, P-bodies, and Cajal bodies [1,2,3,4,5,6,7,8,9]. LLPS biogenesis is suggested to be driven by macromolecular interactions, but the primary mechanism behind the formation of these membrane-less compartments is currently unknown. An understanding of the molecular mechanism behind LLPS is not only important from a theoretical point of view; but may also lead to the development of new drugs for the regulation of LLPS-related processes involved in multiple diseases [3,10].

Currently, LLPS is classified as either associative or segregative [11]. Associative LLPS in aqueous media involves the formation of coacervates, i.e., complexes of oppositely charged polyelectrolytes forming a concentrated phase, which separates from the dilute phase. In systems without coacervates, temperature-dependent associative LLPS can still occur [12,13]. On the other hand, segregative LLPS results in the separation of two (or more) aqueous phases, each enriched in one of the phase-forming compounds in the initial mixture. Our primary focus is on segregative LLPS, although both types of phase separation are known to depend on the physicochemical properties of the solvent [14,15,16,17,18,19,20,21,22].

Aqueous two-phase systems (ATPSs) have been proposed as the simplest in vitro model system for analyzing the governing principles behind the emergence of LLPS [23,24]. ATPS arise in water when the concentrations of two different solutes, typically polymers, such as Dextran or Ficoll™, exceed a certain threshold. The resulting two phases are formed, each containing predominantly one of the two polymers with water constituting up to 80–90 wt.% of the solution [24]. ATPSs can also be formed with only one polymer and either a salt or other low molecular weight additive [24,25,26]. ATPSs formed by two non-ionic polymers and/or salts have been used to separate and analyze a variety of biological macromolecules [27,28,29,30,31]. One necessary condition for phase separation is the emergence of an interface, and thus interfacial tension [31,32,33]. The values of interfacial tension have also been reported to be similar for ATPSs and isolated membrane-less organelles [34,35].

Although the primary molecular mechanism behind phase separation in ATPSs is unknown, a common but incorrect approach is based on the use of the Flory-Huggins theory [36,37]. The original Flory-Huggins theory was developed for polymer mixtures in an organic solvent. This theory is based on the van der Waals model for intermolecular interactions between different polymers in non-polar systems [38]. Although this theory was not meant to represent mixtures of polar compounds, specifically polymers in polar solvents like water, it has been systematically applied to aqueous polymer systems. Multiple arguments against using this model for aqueous polymer mixtures are available in the literature [24,36,37,39,40,41]. Under this model, a proposed mechanism behind LLPS is described using only solute-solute repulsive interactions. However, phase separation is known to depend on the physicochemical properties of the solvent [14,15,16,17,18,19,20,21,22], so solvent-solute interactions cannot be ignored in future models of LLPS.

The solvent properties of water in individual phases of ATPSs have been well characterized using solvatochromic and partitioning studies [16,27,28,29,30]. These characteristics include solvent dipolarity/polarizability (π*) representing solvent dipole–dipole and dipole-induced-dipole interactions, solvent hydrogen bond acceptor basicity (β), hydrogen bond donor acidity (α) and the electrostatic properties of coexisting phases (c). The solvent properties of the aqueous media in the phases of ATPSs are different and depend upon the composition of the phases. The partition behavior of various solutes, including proteins in ATPSs, is therefore governed by the solvent properties of the phases [16,24,27,28,29,42,43]. The interfacial tension of ATPSs has also been established to strongly correlate with the difference between these solvent properties of water in the two coexisting phases [31].

The experimentally observed differences in the solvent properties of water in aqueous solutions have been suggested to originate from the rearrangement of water hydrogen bonds (H-bonds) [42]. The arrangement of H-bonds can be estimated using Attenuated Total Reflection-Fourier Transform Infrared (ATR-FTIR) spectroscopy for analysis of the OH-stretch band [42]. We have previously established that the OH-stretch band may be described with a model based on the decomposition of the OH-stretch band into four Gaussian components assigned to different subpopulations of water [42]. In this approach, the positions of the Gaussians are fixed for pure water, as well as for aqueous solutions of individual compounds. Although the estimated relative contributions of these components depend on the solute type and concentration, the physical dimensions of each subpopulation are currently unknown. Therefore, our assignment of the Gaussian components to water structures is ambiguous. Components at lower optical frequencies are generally assigned to water molecules forming strong, ice-like, H-bonds, while those at higher frequencies are assigned to water molecules in an environment with weaker and/or distorted H-bonds [44]. Fitting the OH-stretch band in water and all the aqueous solutions of various compounds with one, two, three, four, and five Gaussian components showed that the satisfactory fit was always obtained with exactly four components. From the analysis of different assignments of these and other differently positioned components used in the literature [44,45,46,47,48] we have assigned these four subpopulations of water as the following: water with four tetrahedrally arranged H-bonds (3080 cm^−1^), water with four distorted H-bonds (3230 cm^−1^), water with loosely arranged three or four H-bonds (3400 cm^−1^), and water with three, two, or one H-bond(s) (3550 cm^−1^). This assignment is only a rough approximation of the complex H-bond network existing in water [45].

We have established that the relative contributions of these four water subpopulations in aqueous solutions vary significantly, depending on the chemical nature and concentration of a given solute. We have reported results for aqueous solutions of inorganic salts and small organic compounds, such as urea and trimethylamine N-oxide, nonionic polymers, such as polyethylene glycol (PEG), polyvinylpyrrolidone (PVP), and a copolymer of ethylene glycol and propylene glycol (Ucon) [42]. We also reported that the arrangement of H-bonds in the two phases of various ATPSs are different [49]. Recently, we also found that this model of four Gaussian spectral components is applicable to aqueous solutions of globular proteins [50]. We have suggested [42,49,50] that phase separation in aqueous mixtures of two solutes (e.g., different polymers or a single polymer and inorganic salt) occurs due to the formation of dissimilar water micro-domains in the mixture. We hypothesize that these micro-domains increase in both size and dissimilarity with increasing concentration of the phase-forming components prior to macroscopic phase separation. Once these differences reach a specific threshold, interfacial tension emerges where two physical phases are formed. Here, we report the first evidence of the emergence of polymer agglomerates prior to visual macroscopic phase separation via experimental data obtained using dynamic light scattering (DLS). We present these alongside data showing observed rearrangement of H-bonds preceding phase separation in aqueous mixtures of polymers and those of a single polymer and salt using ATR-FTIR spectroscopy.

## 2. Results

### 2.1. Mesoscopic Changes in the Analyzed Systems Detected by DLS

Data obtained using DLS are presented in Figure 1, Figure 2, Figure 3 and Figure 4 as hydrodynamic diameter distribution by volume derived from the intensity profile for agglomerates of individual polymers, or ATPSs mixed and diluted to various degrees. We first measured the size of individual polymers in PBS solutions at concentrations from 1 to 10 wt.% (Figure 1). We estimated from the data in Figure 1a that PEG-8000 has a hydrodynamic radius of 2.29 ± 0.04 nm, which is in good agreement with the estimated literature value of 2.45 nm [51]. From the data in Figure 1b, the hydrodynamic radius of Dextran-75 is estimated as 4.2 ± 0.2 nm—lower than the values reported in the literature, 6.49 nm for Dextran-73 [52]. The difference between estimates may be due to the fact that the literature values were obtained via viscosity measurements of Dextran in water, while ours were obtained via DLS measurements of Dextran in PBS. We also find that our estimates for Ficoll-70 (8.9 ± 0.2 nm) and Ucon-3930 (4.4 ± 1.1 nm), data from Figure 1d,c, respectively, are in good agreement with literature values [53,54].

Data obtained using DLS for dilutions of the mixed ATPS consisting of 11.1 wt.% PEG-8000 and 6.3 wt.% Na_2_SO_4_ are presented in Figure 2. Here, we observe distinct differences in the size of particles in aqueous mixtures of PEG-8000 and Na_2_SO_4_ compared to that of PEG-8000 alone. These data indicate that agglomerates of PEG-8000 are formed and appear to increase in size with increasing concentrations of the polymer and salt preceding visual macroscopic phase separation.

Figure 3 shows the DLS data obtained for various dilutions of ATPSs formed with Dextran-75 and either PEG-8000 (Figure 3a) or Ucon-3930 (Figure 3b), mixed and diluted with PBS to various degrees. In both aqueous mixtures of 8% Dextran-75/5%wt. PEG-8000 and 7.5% Dextran-75/6.2% Ucon-3930, we see a similar trend as observed in the PEG/salt mixtures, where there appears to be an increase in size of agglomerates as the mixtures becomes more concentrated.

Once again, we observe a comparable trend between Ficoll-70 containing mixtures, as in the Dextran-PEG, Dextran-Ucon, and PEG-salt mixtures. Figure 4 shows the DLS data obtained for two ATPSs formed with Ficoll-70 and either PEG-8000 (Figure 4a) or Ucon-3930 (Figure 4b), obtained from the corresponding ATPSs after mixing and dilution with PBS to various degrees.

### 2.2. Changes in Water Structure Detected by ATR-FTIR

Analysis of the OH-stretch spectral band for aqueous mixtures of two polymers obtained by dilutions of a various ATPSs shows that all fractions of the water subpopulations I-IV change with dilution of the mixture, similar to the general trends discussed previously [42,49,50]. An illustrative example of the ATR-FTIR OH-stretch band spectra for 0.15 M NaCl in 0.01 M Na-phosphate buffer, pH 7.4 (PBS) and its decomposition are shown in Appendix A. However, what differs in these mixtures is the appearance of “abrupt changes” in the concentration dependencies rather than the gradual dependencies we observe in solutions of individual polymers. Here, we show three examples (Figure 5, Figure 6 and Figure 7), but the observation of abrupt changes holds true for all 13 systems analyzed in this study. Additional data are presented in Appendix A. The red and blue curves in Figure 5, Figure 6 and Figure 7 represent the effects of individual phase-forming polymers on the H-bond arrangement in their individual solutions in PBS. Comparison of the data obtained in the mixtures to those calculated as a sum of contributions of those determined in the individual solutions of the same polymers show that their effects are clearly nonadditive.

When we compare the polymer concentrations where we first observe abrupt changes in the various dilutions of the ATPS consisting of 8% Dextran-75 and 5% PEG-8000 (Figure 5) to the phase diagram of the system (see Section 4), we detect these abrupt changes before predicted phase separation. The initial part of the concentration dependence of the fraction of water subpopulation I (Figure 5a) of the mixed ATPS dilutions follow a similar trend to that observed for the individual solution of Dextran-75 up until the first abrupt change before the estimated disruption point of the system. Similar coincidences are observed in Figure 7 for fractions of water subpopulations II and III in PEG-8000 and Na_2_SO_4_ mixtures. The abrupt changes in the concentration dependence for the fraction of water subpopulations II and IV are also observed at the same polymer concentrations (Figure 5b,d, respectively).

The data presented in Figure 6 show that dilutions of the ATPS compared to aqueous mixtures of the two individual polymers (Ficoll-70 and Ucon-3930) alone on the H-bond arrangement are also non-additive, similar to the observed for aqueous mixtures of Dextran-75 and PEG-8000.

The data obtained for the PEG-8000/Na_2_SO_4_ ATPS are presented in Figure 7. Behavior of the aqueous mixture of PEG-8000 and Na_2_SO_4_ is surprisingly similar to that of two polymers. Again, we observe the abrupt changes in the concentration dependences of the fractions of water subpopulations I-IV prior to visual phase separation and lack of additivity of the effects of the polymer and salt on H-bonds arrangement in their individual solutions.

We observe similar trends in the FTIR spectrum decompositions for other ATPSs explored here (Table 1). The data for these systems are presented in the Appendix A.

## 3. Discussion

The DLS results show unambiguously that in polymer-polymer and polymer-salt ATPSs, the appearance of polymer agglomerates takes place prior to macroscopic phase separation. This finding strongly supports our previously suggested hypothesis [31] and ATR-FTIR data reported here, that mixtures of two polymers and single polymer and salt form dissimilar water domains filled with phase-forming components which increase in size and dissimilarity before visual phase separation occurs. The observed particles via DLS measurements appear to increase in size with polymers’ (and salt) concentration and are likely to be formed predominantly by PEG in PEG-Na_2_SO_4_, Dextran-PEG, and Ficoll-PEG mixtures and by Ucon in Dextran-Ucon and Ficoll-Ucon mixtures. This is due to the fact that the PEG-rich and Ucon-rich phases both are upper phases in the corresponding ATPSs, i.e., they have lower densities compared to their bottom phases, and it has been well established that the droplets of upper phase are commonly observed floating in the more dense bottom phase [55,56].

The ATR-FTIR data show that the scale of the observed changes in the rearrangement of H-bonds in the studied solutions of individual polymers and aqueous mixtures of polymers is small (in some cases not exceeding ~5%). These changes in the fractional contributions of one or two different water sub-populations, however, correlate strongly with the previously reported experimentally measured solvent features of water (π*, α, and β) [42]. We have already established that our model based on four Gaussian components is both sufficient and necessary to describe the properties of aqueous media in the co-existing phases of ATPSs formed with two polymers and a single polymer and salt [49]. Despite the small changes in relative contributions of each component, these contributions of the individual phases of ATPSs have a linear relationship with the hydrophobic and electrostatic properties of the coexisting phases [49]. Although the scale may appear inconsequential, small changes in the solvent properties of water have been established to correlate strongly with various physicochemical properties of aqueous solutions of different compounds [57]. In Figure 8, we present the results of the DLS measurements for five separate ATPSs (particle size data from Figure 2, Figure 3 and Figure 4), this time with the size of polymer agglomerates as a function of polymer concentration.

Analysis of the data obtained with both DLS and ATR-FTIR for ATPSs are presented in Figure 9 with the size of agglomerates as a function of the fraction of water subpopulation II in the mixtures obtained by dilutions of each ATPS. The observed relationships between the size of polymer agglomerates and the fraction of the water subpopulation II (water in the environment with four distorted H-bonds) are convenient for comparison of the agglomerates’ sizes observed in ATPSs formed by various pairs of components but should not be viewed as a proof of direct relationship between the appearance of agglomerates and rearrangement of hydrogen bonds in the solvent media. These data strongly support our previously suggested [31] hypothesis that H-bond rearrangement occurs in aqueous mixtures of phase-forming polymer(s) at the concentration preceding those for macroscopic phase separation and that polymer agglomerates are formed at same concentration. However, additional more detailed studies will be required for understanding if any relationship between these two processes exist.

The data presented in Figure 9 show that the lowest size of agglomerates observed is in the mixtures of PEG and Na_2_SO_4_ is 500 ± 87.8 nm. This may possibly explain the high speed of phases settling in PEG-salt systems. The agglomerates formed in the mixtures of PEG-8000/Dextran-75 and both Ficoll-containing systems are significantly larger (2408 ± 438 nm and 1978 ± 358 nm, respectively). Agglomerates formed in the mixtures of Dextran-75 and Ucon-3930 are the largest that we observe here—5325 ± 672.5 nm. Because the compositions of these agglomerates are unidentified, the reason of why we observe the largest sizes in Dextran-Ucon mixtures is unknown. This may be due to the differences in individual polymer-polymer interactions, but a more detailed study is necessary to fully understand the reason(s) behind different sizes of agglomerates formed in aqueous mixtures of various pairs of polymers. A zoomed-in plot showing size of agglomerates <500 nm formed in the mixtures of Dextran-75/PEG-8000, PEG-8000/Na2SO4, Dextran-75/Ucon-3930, Ficoll-70/Ucon-3930, and Ficoll-70/PEG-8000 as a function of the fraction of the water subpopulation II in corresponding mixtures is presented in Appendix A.

The experimental data obtained in this study enable us to consider the interactions in the aqueous mixtures of phase forming solutes underlying LLPS in ATPSs. If phase separation first occurs on the mesoscopic level, i.e., the system separates into two distinct pro-phases with different solute concentrations, but phase separation does not occur at the mechanical level, the IR spectra will look like that of macroscopic phase separation. This occurs because although the phases exist as a set of many alternating mesoscopic domains, the contributions of these domains to the IR spectra are additive, provided their size is large enough. Therefore, this scenario can be rationalized in terms of a competition between the local, short-range and long-range interactions in the solvent.

Phase separation in a broad sense occurs due to local interactions, however local interactions do not favor macroscopic separation over the mesoscopic counterpart; in fact, due to the entropy contribution, the mesoscopic phase will always win over the macroscopic counterpart at any finite temperature. However, local interactions provide a contribution to the free energy, which is proportional to the area of the boundary between the phases, so that the thermodynamic equilibrium state, which is achieved at the minimal surface energy, will correspond to the macroscopic separation phase, and mesoscopic separation would not exist at all. On the other hand, if long-range interactions favor inhomogeneity at long distances, homogeneous regions of large size become energetically unfavorable. As a result of a competition between the interfacial tension, separation will start in the IR spectroscopy sense first, and at concentrations, predicted by the theory, based on local interactions. Then, upon increasing concentrations, the surface tension will start competing with the non-local interactions more and more efficiently, thus increasing the typical size of the mesoscopic homogeneous regions, and finally taking over completely, resulting in the mechanical/macroscopic separation.

This qualitative picture can be supported by a quantitative theory, based on the solvent concentration functional for an exemplary two-solute aqueous system, *F* ([*n*_1_(*r*)], [*n*_2_(*r*)]), expressing the free energy of the solvent as a function of the position-dependent concentrations of the solutes, [*n*_1_(*r*)] and [*n*_2_(*r*)] may be written as:(1)$$F\left(\left[n_{1}\left(r\right)\right],\;\left[n_{2}\left(r\right)\right]\right)=F_{l}\;\left(\left[n_{1}\left(r\right)\right],\;\left[n_{2}\left(r\right)\right]\right)+F_{gr}\;\left(\left[n_{1}\left(r\right)\right],\;\left[n_{2}\left(r\right)\right]\right)+F_{nl}\;\left(\left[n_{1}\left(r\right)\right],\;\left[n_{2}\left(r\right)\right]\right),$$
with the three terms describing separable properties.

The first term represents the contribution from local concentration fluctuations, i.e., the dependence of the solvent free energy in a homogenous phase of the solute concentrations:$$F_{l}\;\left(\left[n_{1}\left(r\right)\right],\;\left[n_{2}\left(r\right)\right]\right)=\int dr(f_{0}\left(n_{1}\left(r\right),\;n_{2}\left(r\right)\right)).$$

The second term, has three parts as:$$F_{gr}\;\left(\left[n_{1}\left(r\right)\right],\;\left[n_{2}\left(r\right)\right]\right)=\int dr\left(g\left(n_{1}(r\right),\;n_{2}\left(r\right)\right){\left(\nabla n_{1}\left(r\right)\right)}^{2}+h\left(n_{1}(r\right),\;n_{2}\left(r\right)){\left(\nabla n_{1}\left(r\right)\right)}^{2}+f\left(n_{1}\left(r\right),n_{2}\left(r\right)\right)\left(\nabla n_{1}\left(r\right)\cdot \nabla n_{2}\left(r\right)\right),$$
where *f*_0_, *g*, *h*, and *f* are some local functions of the concentrations of the solutes.

All three terms describe corrections for concentration gradients and are responsible for the interfacial tension. According to the standard notation, ∇(*n*_1_(*r*)) and ∇(*n*_2_(*r*)) are the position-dependent vector gradients of the concentration, whose components are partial derivatives with respect to chosen coordinates. The surface tension coefficient is obtained by solving a quasi-one-dimensional optimization problem for the functional containing both local and gradient terms, with the solute concentrations depending on a single coordinate, transverse to the interphase surface. The boundary conditions are given by the solute concentrations at ±∞, reproducing their counterparts in the homogeneous phases.

In its simplest form, the final term, *F_nl_*, of the functional describes long-range interactions dependent upon values of *K_jk_* as:$$F_{nl}\;\left(\left[n_{1}\left(r\right)\right],\;\left[n_{2}\left(r\right)\right]\right)=\int drdr^{\prime }\sum K_{jk}\left(r,\;r^{\prime }\right)n_{j}\left(r\right)n_{k}\left(r^{\prime }\right)).$$

The function *K_jk_* that depends on two positions describes long-range correlations in the dependence of the free energy on the solute concentrations and hence describes specifically the mechanisms of how solute properties change the long-range order of the H-bond network in pure water.

Despite the deceptive simplicity of the three elements of the first equation, this solvent concentration functional likely describes many of the complex dynamical mechanisms driving phase separation, including direct interactions of the solutes, water-solute interactions, and modification of water properties by the solute, including dipole interactions, and donor and acceptor properties of the H-bonds.

Homogeneous solutions of the variational problem for our proposed functional, *F*, using a mean-field approach, but omitting the final long-range term, readily reproduces standard phase-separation diagrams. The gradient correction terms offer a modeling tool for interfacial tension and applying fluctuation theory to the solvent density functional allows metastable configurations to be analyzed. Since interfacial tension tends to minimize the area of an interface, a functional without the nonlocal term can predict only two extreme scenarios of no separation and fully macroscopic separation. The nonlocal term tends to increase the free energy of homogeneous configurations, so that formation of mesoscopic domains of the two phases will minimize the free energy to reach thermodynamic equilibrium. The domain sizes are determined by the competition between non-local and gradient contributions that tend to suppress homogeneity and minimize the interfacial area by decreasing and increasing the domain sizes, respectively. Therefore, between extreme cases of microscopically homogeneous phases, presuming no separation, and macroscopic separation, we are now able to examine the phenomenology of mesoscopic domain formation arising from the relative magnitudes of, and hence competition between, local, long-range, and gradient/surface terms.

All experimental data obtained in this study provide evidence of an abrupt rearrangement of H-bonds in aqueous mixtures of two phase-forming polymers or a single polymer and inorganic salt prior to phase separation. These data confirm the previously suggested hypothesis [49] that the two polymers form different polymer-specific water H-bond network domains, which have dissimilar solvent properties. The dissimilarity between the domains increases with increasing polymer concentrations in the mixture. There are two types of domains that exist in the polymer mixtures until the polymer concentrations exceed a threshold beyond which the domains become immiscible, at which point the emerging interfacial tension leads to the formation of micro-droplets, eventually coalescing into separate layers controlled by the density of the phases. The above data provide evidence that this hypothesis may be extended to the mixtures of a single polymer and inorganic salt.

Importantly, our observations are also supported by the results of the analysis of the effects of hydrogen bond-dependent interactions on the occurrence of microheterogeneity in the aqueous solutions of small polar organic compounds (such as glycerol, ethylene glycol, acetone, tetramethylurea, isopropyl alcohol, tetrahydrofuran, and tert-butanol), as they also show the appearance of the mesoscopic droplets at low concentrations of organic compounds in aqueous solutions [58,59,60]. In fact, the authors demonstrated “spontaneous occurrence of stable mesoscopic heterogeneity with scales on the order of 10–10^3^ nm in aqueous solutions of some organic materials with molecular concentrations (mole fractions) ranging from 10^−1^ to 10^−6^” [60]. Although the analyzed polar organic compounds showed different potential for mesophase separation, which was weak for glycerol, ethylene glycol, and acetone, intermediate for tetramethylurea and isopropyl alcohol, and intense for tetrahydrofuran and tert-butanol, all of them are able to interact with water via H-bonds, most serving solely as H-bond acceptors and some acting both as H-bond acceptors and donors [60]. The authors also emphasized that the amphiphilic structure of these molecules may play a crucial role in the nucleation of the observed mesodroplets [60]. Furthermore, these studies revealed that mesophilic liquid droplets observed in the aqueous solutions of the polar organic compounds are enriched in the dissolved substances, as the concentration of these dissolved organic molecules in droplets was significantly higher than in the bulk solution [58,59,60]. Therefore, formation of the solute-enriched mesoscopic droplets near the spinodal via the H-bond distortion represents a common phenomenon for the aqueous solutions of polar compounds, whether small organic compounds or polar polymers of different nature.

## 4. Materials and Methods

### 4.1. Materials

Polyethylene glycol with an average molecular weight of 8000 (PEG-8000, lot SLBW6815) was purchased from Sigma-Aldrich (St. Louis, MO, USA). Ucon 50-HB-5100, a random copolymer of 50% ethylene oxide and 50% propylene oxide, with molecular weight of 3930 (Ucon-3930, lot SJ1955S3D2), was purchased from Dow-Chemical (Midland, MI, USA). Dextran with an average molecular weight of 75,000 (Dextran-75, lot 119945) was purchased from USB Corporation (Cleveland, OH, USA). Ficoll^TM^ with an average molecular weight of 70,000 (Ficoll-70, lot 10297022) was purchased from Cytiva (Marlborough, MA, USA). Sodium sulfate (Na_2_SO_4_, lot 195740), sodium chloride (NaCl, lot 212436), sodium phosphate dibasic heptahydrate (Na_2_HPO_4_·7H_2_O, lot 190833), and sodium phosphate monobasic monohydrate (NaH_2_PO_4_·H_2_O, lot 194198) of analytical reagent grade were purchased from Fisher Scientific (Waltham, MA, USA). Ultrapure water (18.5 mΩ) purified using a Milli-Q^®^ lab water system (EMD Millipore Sigma, Burlington, MA, USA) was used for the preparation of all solutions.

### 4.2. Methods

#### 4.2.1. Preparation of Aqueous Two-Phase Systems

Stock solutions of PEG-8000 (50 wt.%), Ucon-3930 (60 wt.%), Dextran-75 (42.1 wt.%), Ficoll-70 (46.2 wt.%), and Na_2_SO_4_ (20.3 wt.%) were prepared in water. A stock solution of 3 M NaCl in 0.2 M sodium phosphate buffer, pH 7.4 was prepared by mixing 35.07 g of NaCl, 8.69 g of Na_2_HPO_4_·7H_2_O, and 1.05 g of NaH_2_PO_4_·H_2_O in 200 mL of water. A separate stock solution of 0.15 M NaCl in 0.01 M Na-phosphate buffer, pH 7.4 (PBS) was prepared and used as the diluent in all experiments conducted in this study.

ATPSs with the compositions listed in Table 1 were prepared for ATR-FTIR analysis by adding appropriate amounts of the aqueous stock solutions for PEG-8000, Ucon-3930, Dextran-75, Ficoll-70, and/or Na_2_SO_4_ into a 5 mL transport tube. Appropriate amounts of 3.0 M NaCl in 0.2 M Na-phosphate buffer, pH 7.4 were added so as to give the required ionic composition. Water was then added to obtain a 4 g final system. Solutions were mixed vigorously and allowed to settle at ambient temperature (~23 °C) to determine if ATPSs would form. Once an ATPS was established, several identical mixtures of two polymers (or PEG-8000 and Na_2_SO_4_) were prepared for each ATPS examined. For each set of ATR-FTIR or DLS experiments, each ATPS was vigorously mixed, quickly aliquoted and each aliquot was diluted with PBS, pH 7.4 to ~50, 60, 70, 80, 90, 95, and/or 97 wt.% of the initial mixture. Actual wt.% varied from these exact values and were calculated appropriately. Phase diagrams for the systems probed using DLS are listed in Figure 10. For each system, the line of dilution is shown connecting each specific ATPS to the origin. Where this line crosses the binodal curve, we refer to as the disruption point for that system, i.e., where we no longer observe two distinct phases visually.

#### 4.2.2. Attenuated Total Reflection—Fourier Transform Infrared Spectroscopy Analysis

ATR-FTIR spectra for each sample were measured in two separately prepared solutions using an Alpha II FT-IR spectrometer (Bruker) equipped with Platinum single reflection ATR single reflection diamond ATR module (Bruker Scientific, LLC, Billerica, MA, USA). An illustrative example of the ATR-FTIR spectra of OH-stretch band in the solvent for 0.15 M NaCl in 0.01 M Na-phosphate buffer, pH 7.4 (PBS) and its decomposition are shown in Appendix A. All measurements were performed at ambient temperature (23 °C) using 24 scans for each sample and 24 scans for background in the spectral range of 4000–1000 cm^−1^ with resolution of 4 cm^−1^. The spectra were reproducible to better than 1 cm^−1^. Some of experiments for the indicated polymer mixtures and systems were performed at Aveiro University using Spectrum Bx FT-IR spectrometer (Perkin-Elmer, Waltham, MA, USA) equipped with single reflection Golden Gate diamond ATR (Specac). All measurements were performed at 25 °C using 20 scans for each sample, 24 scans for background in the spectral range of 4000–1000 cm^−1^ with resolution of 2 cm^−1^. The spectra were reproducible to within ±1 cm^−1^. Analysis of the OH-stretch band obtained with these measurements was performed and compared with those obtained in the Cleveland Diagnostics laboratory with the Bruker ATR-FTIR spectrometer. The replicability of the data was found to be better than 0.2%.

The ATR module is placed in the sample compartment. The spectrometer is located in the thermostat enclosure with the temperature maintained at 23 ± 1 °C. The agreement between the data obtained in our laboratory and in the laboratory in Aveiro University using different ATR-FTIR spectrometer shows that the data are highly reproducible within about a 2 °C range. It should also be mentioned that typically the ATR-FTIR spectra were measured for not more than 20 samples at a time (within less than 1 hr.). Within this time period the ATR module temperature monitored with the thermosensor was maintained within the indicated interval.

ATR-FTIR spectra were analyzed using custom software written in Wolfram Mathematica and run under version 12. Details on the software and protocol used can be found in [50].

#### 4.2.3. Dynamic Light Scattering

Hydrodynamic size and size distributions were estimated by dynamic light scattering (DLS, ZS Zetasizer Nano ZSP, Malvern Panalytical, Malvern, UK) with a He-Ne laser (633 nm, 4 mW) as a light source. Analysis was performed using the light scattering software DTS application. Default values for aqueous solutions (Dispersant name: water; Dispersant RI: 1.330; Viscosity (cP): 0.8872) were used in the DTS analysis. The scattering light was collected at a 173° backscattering angle and a wavelength of 633 nm at ambient temperature (~25 °C). Ten scans of 10 s each were performed at the studied conditions. The reported values are an average of at least six measurements. Note that for the aqueous polymer mixtures at the concentration below binodal line, there is no phase separation, hence the mixtures were not mixed between the repeated measurements.

Since the amount of scattered light from the particles depends on the relative refractive index, to be detectable by DLS, the refractive index (RI) of particles should be different from that of the RI of dispersant. Since in the DLS experiments, accurate estimation of the dimensions of particles depend on the viscosity and RI of the solution, the RI and viscosity of dispersant should be known with accuracy better than 0.5%. Obviously, the viscosity and refractive index of polymer solutions are very different from those of pure water, depending on the concentrations of the polymers as well as its nature and molecular mass of the dissolved polymers, with viscosity being also dependent on the solution temperature.

The viscosity of PEG-Na_2_SO_4_ ATPS is close to that of water, and concentration of PEG did not exceed 10 wt.% at the highest polymer concentration. Since this system contains only one polymer (PEG) the RI does not change, and the data for individual PEG solutions obtained with concentration range up to 50 wt.% showed the hydrodynamic diameter of PEG in very good agreement with the literature data. The viscosity of a 10 wt.% solution of PEG-8000 exceeds that of water by approximately 2% [61]. Hence, the data obtained for mixtures of PEG and Na_2_SO_4_ seem to provide the correct size estimates of agglomerates.

One should keep in mind that systems analyzed in our study are very different from the traditional subjects of the DLS analysis, which represent solutions containing well-defined particles dispersed within the well-defined and invariant solvent. In our case, dispersants represent polymer solutions with different concentrations of the polymers and other solutes, whereas the observed “particles” represent the agglomerates of the polymers already present in the dispersants. Taking into account the complexity of these systems, providing accurate estimates of the RI and viscosity of dispersants is challenging. Therefore, in our DLS analysis, we used default values for aqueous solutions (dispersant RI of 1.330 and viscosity of about 0.9 mPa × s) even though the actual viscosity and RI of the dispersants used in this study are different from those of pure water. As a result, the estimated “hydrodynamic diameters” of the mesoscopic polymer agglomerates observed in this work admittedly represent very crude estimates. Although the accurate evaluation of the actual dimensions of these mesoscopic polymer agglomerates undoubtedly represents an important task, such research is outside the goal of the present work, which aims to provide some qualitative evidence that in the systems undergoing liquid-liquid phase separation (such as aqueous mixtures of two polymers and a single polymer and salt), the mesoscopic changes precede visually detected macroscopic phase separation. More quantitative analysis of this phenomenon represents important future development, and the corresponding studies are currently underway in our laboratories.

## 5. Conclusions

The data obtained in this study using dynamic light scattering show that as previously predicted [31], mesoscopic agglomerates of polymer appear in aqueous solutions of two polymers or a single polymer and inorganic salt at concentrations significantly lower than those corresponding to visual macroscopic phase separation. With increasing polymer concentrations in the solution, the size of these agglomerates increases. The size of agglomerates and their concentration dependence varies with the chemical nature of the phase-forming polymers.

Analysis of H-bond arrangement in aqueous solutions of two polymers or a single polymer and inorganic salt performed by examination of the OH-stretch band in the ATR-FTIR spectra showed that the formation of polymer agglomerates is accompanied by abrupt changes in the fraction(s) of one or more subpopulation(s) of water assigned to water molecules in the environment of water molecules with four tetrahedrally arranged H-bonds (subpopulation I), water molecules in an environment with four distorted H-bonds (subpopulation II), water with loosely arranged three or four H-bonds (subpopulation III), and/or water with three, two, or one H-bond (s) (subpopulation IV). The relationships between fraction of water subpopulation II and the size of polymer agglomerates described by power of exponential function were observed. Additional analysis of the agglomerates, their size, and its changes in terms of rearrangement of hydrogen bonds in various aqueous two-phase systems must be examined in further detail.

An initial theoretical model of interactions responsible for the mechanism of phase separation in aqueous mixtures of two polymers or a single polymer and inorganic salt is proposed, and further development of the model is in progress. It may be suggested that the underlying principles that underly mechanism of phase separation in aqueous two-phase systems are similar to those governing liquid-liquid phase separation in aqueous media and formation of membrane-less organelles (biocondensates), in particular. In summary, the following conclusions can be made:Polymer agglomerates are formed in the aqueous mixtures of two polymers and single polymer and salt prior to visual macroscopic phase separation.Abrupt changes in the H-bond rearrangement in aqueous mixtures of two phase-forming polymers or a single polymer and inorganic salt occurs prior to the visually detectable macroscopic phase separation. This finding implies that the separation of mesoscopic pro-phases represents the initial state in the formation of aqueous two-phase systems.A theoretical model of liquid-liquid phase separation in aqueous media, based on the solvent concentration functional, is supported by these experimental findings.

## Figures and Tables

**Figure 1 ijms-23-14366-f001:**
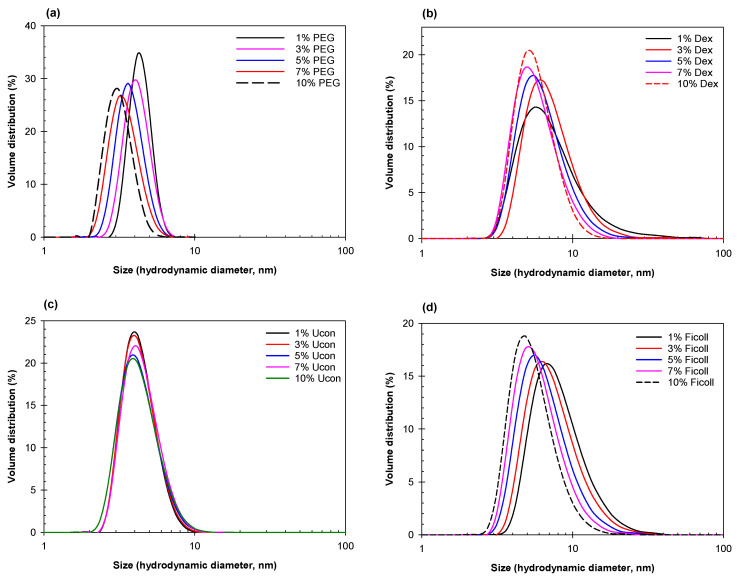
Size distribution for (**a**) PEG-8000, (**b**) Dextran-75, (**c**) Ucon-3930, and (**d**) Ficoll-70 solutions at different concentrations indicated. All solutions prepared in PBS; concentrations in weight %.

**Figure 2 ijms-23-14366-f002:**
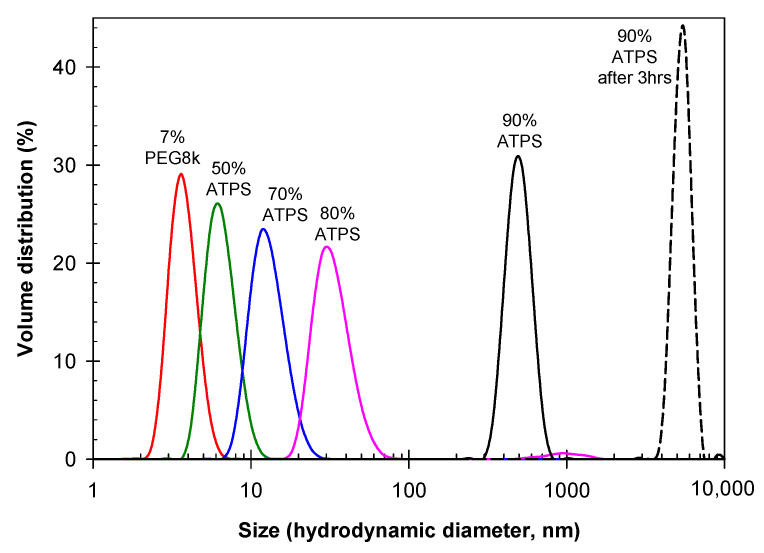
Size distribution for 7% PEG-8000 (red) compared to various dilutions of an ATPS consisting of 11.1% PEG-8000 and 6.3% Na_2_SO_4_. Mixed dilutions of the ATPS correspond to 5.55% PEG-8000 and 3.15% Na_2_SO_4_ (green), 7.77% PEG-8000 and 4.41% Na_2_SO_4_ (blue), 8.88% PEG-8000 and 5.04% Na_2_SO_4_ (pink), 10% PEG-8000 and 5.7% Na_2_SO_4_ freshly mixed (black), and after standing for 3 h (black dashed). All solutions prepared in PBS; concentrations in weight %.

**Figure 3 ijms-23-14366-f003:**
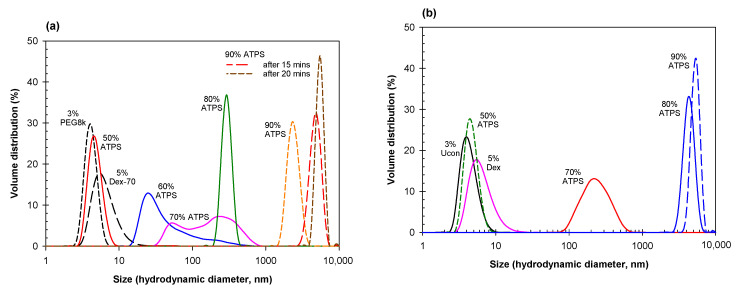
(**a**) Size distribution for 3% PEG-8000 (short black dash) and 5% Dextran-75 (long black dash) compared to various dilutions of an ATPS consisting of 8% Dextran-75 and 5% PEG-8000. Mixed dilutions of the ATPS correspond to 4.0% Dextran-75 and 2.5% PEG-8000 (red), 4.8% Dextran-75 and 3.0% PEG-8000 (blue), 5.6% Dextran-75 and 3.5% PEG-8000 (pink), 6.4% Dextran-75 and 4.0% PEG-8000 (green), along with 7.2% Dextran-75 and 4.5% PEG-8000 freshly mixed (orange dashed), after standing for 15 min (red dashed), and after standing for 20 min (brown dashed). (**b**) Size distribution for 3% Ucon-3930 (black) and 5% Dextran-75 (pink) compared to various dilutions of an ATPS consisting of 7.5% Dextran-75 and 6.2% Ucon-3930. Mixed dilutions of the ATPS correspond to 3.75% Dextran-75 and 3.1% Ucon-3930 (green dashed), 5.25% Dextran-75 and 4.34% Ucon-3930 (red), 6.0% Dextran-75 and 4.96% Ucon-3930 (blue), and 6.75% Dextran-75 and 5.58% Ucon-3930. All solutions prepared in PBS; concentrations in weight %.

**Figure 4 ijms-23-14366-f004:**
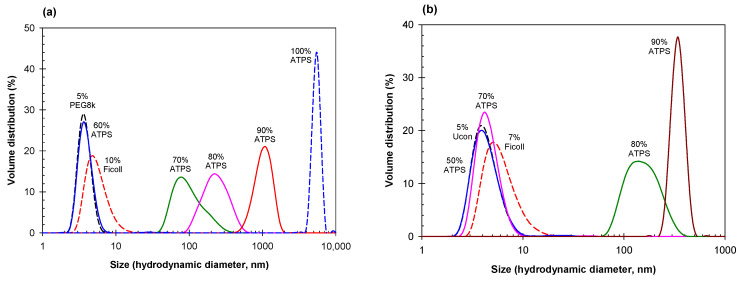
(**a**) Size distribution for 5% PEG-8000 (short black dash) and 10% Ficoll-70 (short red dash) compared to various dilutions of an ATPS consisting of 17% Ficoll-70 and 8% PEG-8000. Mixed dilutions of the ATPS correspond to 4.8% PEG-8000 and 10.2% Ficoll-70 (blue), 5.6% PEG-8000 and 11.9% Ficoll-70 (dark green), 6.4% PEG-8000 and 13.6% Ficoll-70 (pink), 7.2% PEG-8000 and 15.3% Ficoll-70 (red), and 8% PEG-8000 and 17% Ficoll-70. (**b**) Size distribution for 5% Ucon-3930 (short black dash) and 7% Ficoll-70 (short red dash) compared to various dilutions of an ATPS consisting of 12.5% Ficoll-70 and 7.6% Ucon-3930. Mixed dilutions of the ATPS correspond to 4.5% Ucon-3930 and 6.75% Ficoll-70 (blue), 6.3% Ucon-3930 and 9.45% Ficoll-70 (pink), 7.2% PEG-8000 and 10.8% Ficoll-70 (dark green), and 8.1% Ucon-3930 and 12.15% Ficoll (brown). All solutions prepared in PBS; concentrations in weight %.

**Figure 5 ijms-23-14366-f005:**
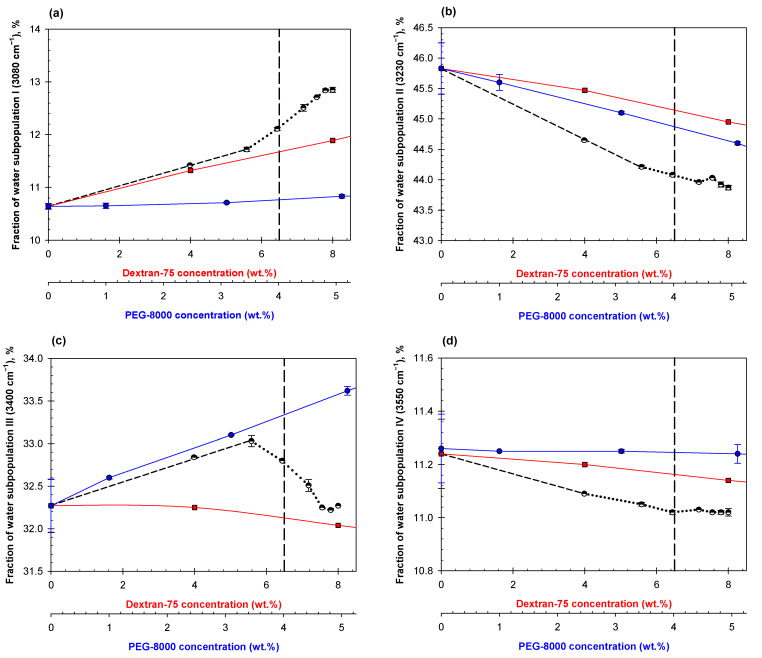
Concentration dependences of the water subpopulation fractions represented by the relative contribution of Gaussian components I (**a**), II (**b**), III (**c**), IV (**d**) on dilutions of mixed ATPS in PBS consisting of 8% Dextran/5% PEG (black connected by a dashed line until abrupt change occurs then dotted black line), Dextran-75 in PBS (red squares), and PEG-8000 in PBS (blue circles). The vertical dashed line represents the estimated disruption point of this system according to its phase diagram.

**Figure 6 ijms-23-14366-f006:**
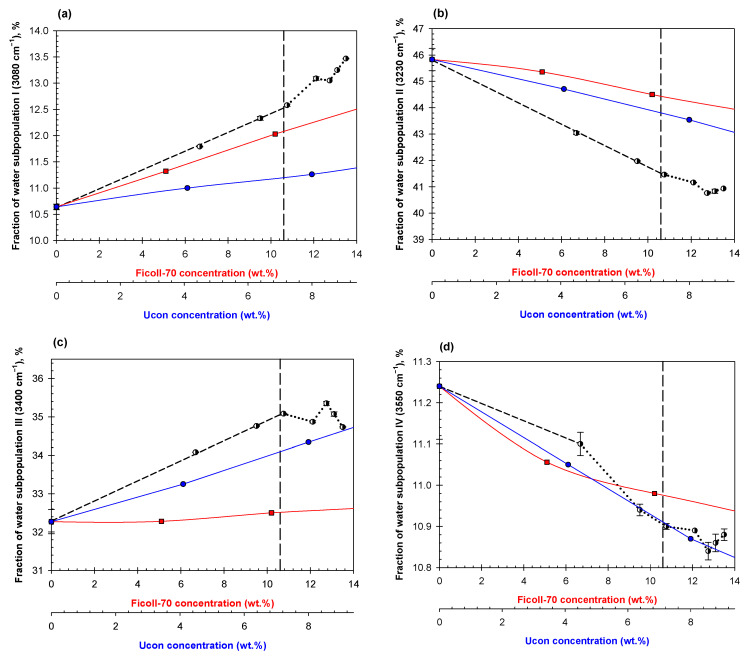
Concentration dependencies of the relative contribution of Gaussian components I (**a**), II (**b**), III (**c**), IV (**d**) on dilutions of mixed ATPS in PBS consisting of 13.5% Ficoll-70/9% Ucon-3930 (black dashed line until abrupt change occurs then dotted black line), Ficoll-70 in PBS (red squares), and Ucon-3930 in PBS (blue circles). The vertical dashed line represents the estimated disruption point of this system according to its phase diagram.

**Figure 7 ijms-23-14366-f007:**
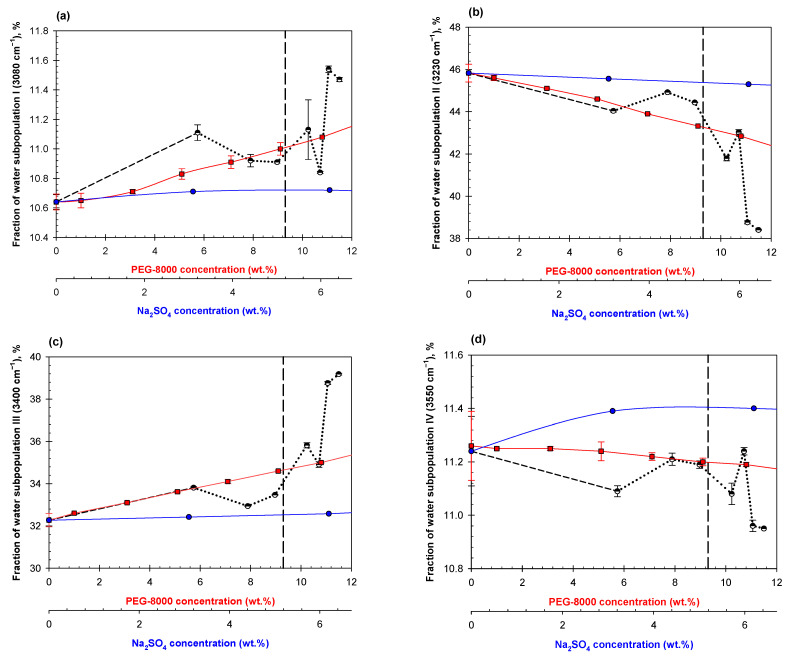
Concentration dependencies of the relative contribution of Gaussian components I (**a**), II (**b**), III (**c**), IV (**d**) on dilutions of mixed ATPS in PBS consisting of 11.1% PEG/6.3% Na_2_SO_4_ (black dashed line until abrupt change occurs then dotted black line), PEG-8000 in PBS (red squares), and Na_2_SO_4_ in PBS (blue circles). The vertical dashed line represents the estimated disruption point of this system according to its phase diagram.

**Figure 8 ijms-23-14366-f008:**
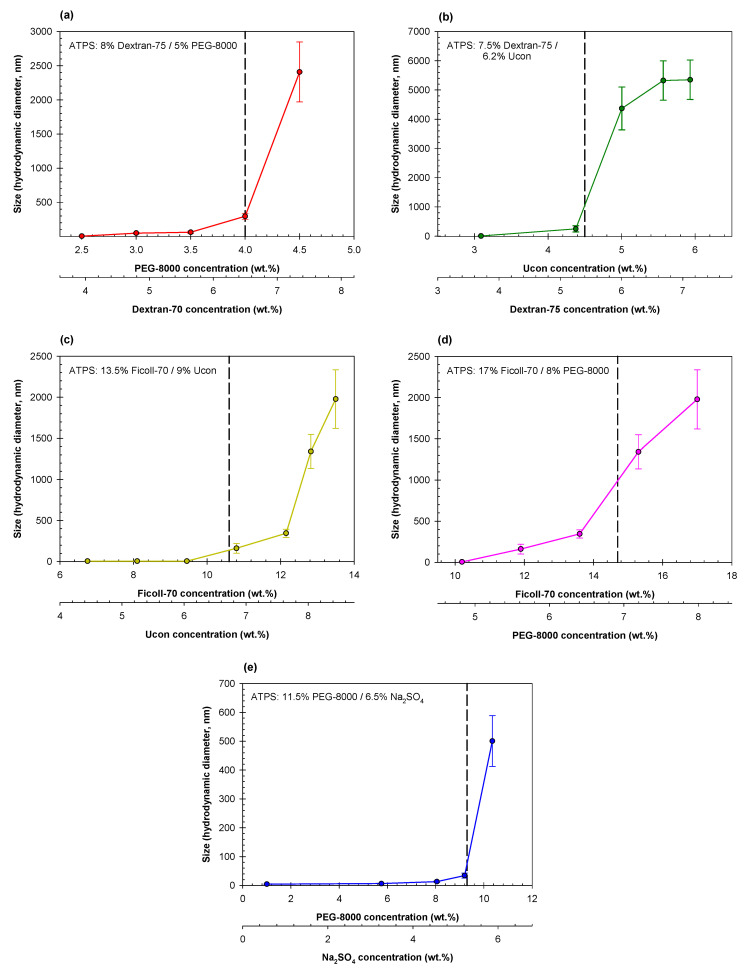
Size of polymer agglomerates as a function of polymer concentration in ATPSs formed by: (**a**) Dextran-75/PEG-8000 (red), (**b**) Dextran-75/Ucon-3930 (green), (**c**) Ficoll-70/Ucon-3930 (yellow), (**d**) Ficoll-70/PEG-8000 (pink), and (**e**) PEG-8000/Na_2_SO_4_ (blue).

**Figure 9 ijms-23-14366-f009:**
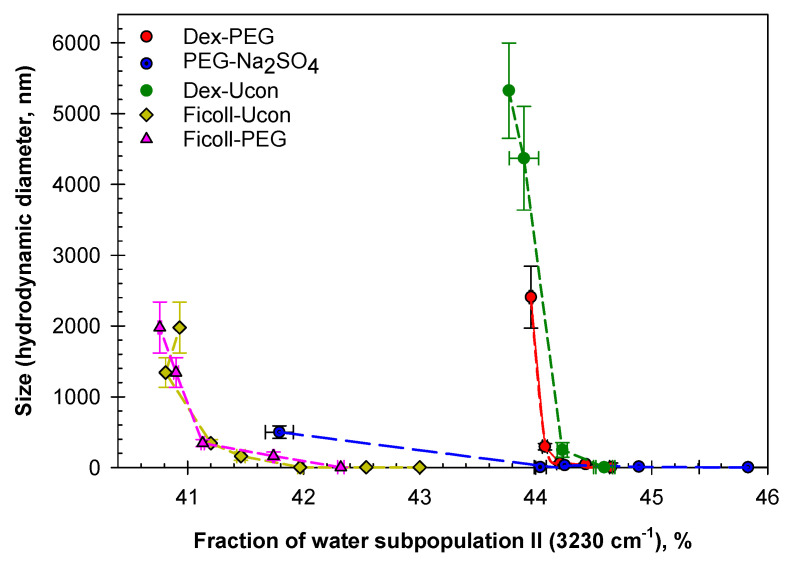
Size of agglomerates formed in the mixtures of Dextran-75/PEG-8000 (red), PEG-8000/Na_2_SO_4_ (blue), Dextran-75/Ucon-3930 (dark green), Ficoll-70/Ucon-3930 (yellow), and Ficoll-70/PEG-8000 (pink) as a function of the fraction of the water subpopulation II in the mixtures.

**Figure 10 ijms-23-14366-f010:**
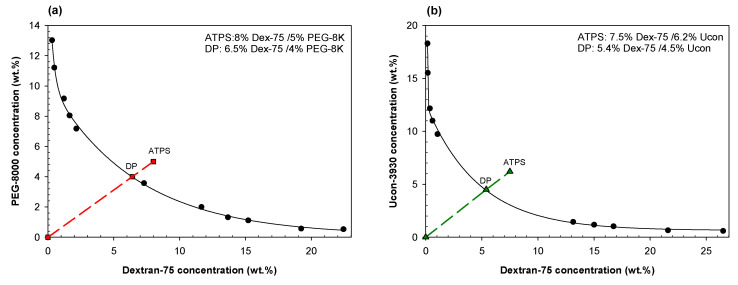
Phase diagrams of (**a**) Dextran-75/PEG-8000, (**b**) Dextran-75/Ucon-3930, (**c**) Ficoll-70/Ucon-3930, (**d**) Ficoll-70/PEG-8000, and (**e**) PEG-8000/Na_2_SO_4_. Data points for binodals in a-d were adapted with permission from [49]. The five system compositions (ATPS) used here are those that were measured using DLS and are plotted along their lines of dilution (dashed lines) with their estimated disruption points (DP) on the binodal.

**Table 1 ijms-23-14366-t001:** Polymer compositions of whole systems and their estimated disruption points. All ATPSs contained a total ionic composition of 0.15 M NaCl in 0.01 Na-phosphate buffer, pH 7.4.

Polymer 1	Polymer 2/ Salt	Total Composition (wt.%)		Disruption Point (wt.%)	
		Polymer 1	Polymer 2/ Salt	Polymer 1	Polymer 2/ Salt
PEG-8000	Dextran-75	5.2	5.0	5.0	4.8
PEG-8000	Dextran-75	3.8	7.6	3.6	7.2
PEG-8000	Dextran-75	5.0	8.0	4.0	6.5
PEG-8000	Dextran-75	2.8	10	2.6	9.3
Ucon-3930	Dextran-75	6.2	7.5	4.5	5.4
Ucon-3930	Dextran-75	3.0	10	2.6	8.5
Ficoll-70	Ucon-3930	13.5	9.0	10.6	7.1
Ficoll-70	Ucon-3930	7.6	12.5	6.5	10.8
PEG-8000	Ficoll-70	8.0	17	6.9	14.7
PEG-8000	Ficoll-70	7.8	14.8	7.4	13.9
PEG-8000	Na_2_SO_4_	5.1	7.2	4.9	6.9
PEG-8000	Na_2_SO_4_	11.1	6.3	9.4	5.3
PEG-8000	Na_2_SO_4_	14	4.5	13.3	4.3
PEG-8000	Dextran-75	5.2	5.0	5.0	4.8

## Data Availability

All raw DLS and ATR-FTIR data along with the Wolfram Mathematica code used are available upon request.

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
