# Peer review of "Mechanism of Phase Separation in Aqueous Two-Phase Systems"

_ijms, 2022, doi:10.3390/ijms232214366_

Round 1

Reviewer 1 Report

Comments for the author.

In the Figure 1 the plot (c) is different, it is not clear why. What do the color dashed lines mean?

In the paragraph 2.2.2 authors are describing ATR-FTIR measurements but no spectra are shown. They should publish the raw data at least in the supplementary materials.

In the paragraph 2.2.3 the authors write that at least six measurements were carried out. They should be specify that. Were the systems mixed between measurements?

In the Figure 2 the line for 10% PEG in the plot (a) is almost invisible as well as in the Figure 4a – line after 20 min, the authors should use some more pronounced lines.

Some axes captions in the figures 6,7,8 are cut.

The authors write on the page 9 that: “The concentration dependences of the fractions of water subpopulations I (Figure 6a) are identical to those for the individual solution of Dextran-75 until the abrupt changes in the concentration dependences for the mixtures occurring at about the same polymers concentrations.” This opinion is not obvious.

They write: “The abrupt changes in the concentration dependence for the fraction of water subpopulations II and III are also observed at the same polymer concentrations (Figure 6b and 6d, respectively).” But the subpopulations II and III are shown in the Figures 6b and 6c, respectively

On the same page 9 it is not clear Why “For the concentration dependence of the fraction of water subpopulation III (Figure 6c) the abrupt change is observed at higher concentrations of the polymers though also prior to the phase separation threshold.” In the same Figure 6d: why are not the points fitted by a curve?

The Figure 7 is not described in the main text. In the figure 7d is not clear why all the points are not fit by a trendline. The same problem is in the Figure 8a and b.

In the Figure 9 it is not clear from which concentration data did the authors assume.

On the page 12 the authors write that ATR-FTIR data show … but no spectra are published. The spectra and the fitting process should be shown at least in the supplementary material.

Figure 10: It would be beneficial to show region below ca 500 nm in detail.

On the page 13 down, it is not clear what the “500 ? 87.8 nm” means.

In the conclusion the opinion in the point 3 is not sufficiently substantiated.

Reviewer 2 Report

1. The authors refer to one of their works, where they developed a method for analyzing the distribution of water molecules in a solution according to the degree of hydrogen bonding. At the same time, they use a specific ATR module. I admit that the work was done without errors, but this method is not so general that it can be referred to as an established methodology. Moreover, in the last few decades, similar methods of decomposition of the stretching vibrational band of water have been described in a more general form – for transmission spectra. The difficulty is that it is possible to repeat the experiments of the authors only with the same spectrometer and the same ATR module, since the position of the bands in the ATR spectrum depends on the parameters of this module. I would like to know what is the angle of incidence of radiation on the sample surface in the ATR diamond module used? Can this be extracted from the specification of ATR module?

2. It is known that for aqueous solutions, there is a strong displacement of intramolecular bands with temperature changes, therefore, conducting an IR analysis at an unstable temperature (as the authors write about 23C) does not look very professional. The authors need to justify that a change in the temperature of the solution by at least 2 degrees C does not lead to changes in the parameters of the decomposition of the IR band comparable to the differences discussed between the solutions. The fact is that in the spectrometer, during operation for several hours, the temperature of the sample compartment may rise by several degrees C. Do I understand correctly that the used ATR module is placed in the sample compartment?

3. Methods of atmospheric suppression in the IR spectrometer are not described. And this is important because the band of liquid water overlaps strongly with the lines of water vapor. During the experiment, was the spectrometer vacuumed, purged with dry air, or was a mathematical correction of water vapor performed?

4. What values of viscosity and refractive index of solutions were set in the DTS software?

5. For the equations on page 14, not all the designations of the parameters are given.

6. The authors write about the influence of donor and acceptor hydrogen bonds between molecules on the processes of phase separation, but there are practically no references to modern works where these processes were considered, including at the theoretical level. These are, for example, the following works:

http://dx.doi.org/10.1063/1.4966187

https://link.springer.com/article/10.3103/S1541308X18010041

https://link.springer.com/article/10.3103/S1541308X19020031

7. The work obviously belongs to the section of physical chemistry, and may be of interest to researchers of this profile. At the same time, the description of membrane-less organelles in the Introduction and Abstract seems irrelevant to the topic. There is no reason to attribute the results of this work to cell biology, biochemistry or biophysics.

I believe that the work cannot be published in its present form in such an honorable journal as IJMS. However, after taking into account the indicated comments and significant improvements, it has a chance of compliance.

Round 2

Reviewer 1 Report

The authors reacted adequately on all suggestions and comments

Author Response

We thank the reviewer for their comments.

Reviewer 2 Report

My next remark relates to paragraph 4 of report 1 of the review.

4. What values of viscosity and refractive index of solutions were set in the DTS software?

the authors' response

Default values for aqueous solutions were used:

Dispersant name: water; Dispersant RI: 1.330; Viscosity (cP): 0.8872. Corresponding information is added to the revised manuscript.

This is fundamentally wrong. Since the dimensions determined by the DLS method depend on the viscosity of the solution linearly and quadratically on the refractive index. And the viscosity and refractive index of polymer solutions are very different from the parameters for water and change with concentration. In this case, the "hydrodynamic diameters" defined by the authors are not such, without proper consideration of viscosity and refractive index. For example, if you set a viscosity of 0.9 sP (for water) in the program, but in fact it is 1.8 sP, then the determined size will be overestimated by 2 times. Well, if the real viscosity of the solution is 9 sP, then - 10 times!

But the concentrations of polymers used are high. This must be taken into account, otherwise the size data is incorrect.

Round 3

Reviewer 2 Report

My main comments have been taken into account. I believe that the manuscript can be accepted for publication.